# Mesoporous Silicon Nanoparticles with Liver-Targeting and pH-Response-Release Function Are Used for Targeted Drug Delivery in Liver Cancer Treatment

**DOI:** 10.3390/ijms25052525

**Published:** 2024-02-21

**Authors:** Jintao Wei, Yue Tan, Yan Bai, Jincan He, Hua Cao, Jiao Guo, Zhengquan Su

**Affiliations:** 1Guangdong Provincial University Engineering Technology Research Center of Natural Products and Drugs, Guangdong Pharmaceutical University, Guangzhou 510006, China; startwjt@163.com (J.W.); tanyue2024@163.com (Y.T.); 2Guangdong Metabolic Disease Research Center of Integrated Chinese and Western Medicine, Guangdong Pharmaceutical University, Guangzhou 510006, China; 3School of Public Health, Guangdong Pharmaceutical University, Guangzhou 510310, China; angell_bai@163.com (Y.B.);; 4School of Chemistry and Chemical Engineering, Guangdong Pharmaceutical University, Zhongshan 528458, China; caohua@gdpu.edu.cn

**Keywords:** aspirin, mesoporous silica, polydopamine, galactosamine, HepG2 cells

## Abstract

This article aims to develop an aspirin-loaded double-modified nano-delivery system for the treatment of hepatocellular carcinoma. In this paper, mesoporous silica nanoparticles (MSN) were prepared by the “one-pot two-phase layering method”, and polydopamine (PDA) was formed by the self-polymerization of dopamine as a pH-sensitive coating. Gal-modified PDA-modified nanoparticles (Gal-PDA-MSN) were synthesized by linking galactosamine (Gal) with actively targeted galactosamine (Gal) to PDA-coated MSN by a Michael addition reaction. The size, particle size distribution, surface morphology, BET surface area, mesoporous size, and pore volume of the prepared nanoparticles were characterized, and their drug load and drug release behavior in vitro were investigated. Gal-PDA-MSN is pH sensitive and targeted. MSN@Asp is different from the release curves of PDA-MSN@Asp and Gal-PDA-MSN@Asp, the drug release of PDA-MSN@Asp and Gal-PDA-MSN@Asp accelerates with increasing acidity. In vitro experiments showed that the toxicity and inhibitory effects of the three nanodrugs on human liver cancer HepG2 cells were higher than those of free Asp. This drug delivery system facilitates controlled release and targeted therapy.

## 1. Introduction

Nanodelivery systems are a promising in the biomedical field. Compared to traditional drug carriers, nanocarriers improve the pharmacokinetics and biodistribution of therapeutic drugs, thereby maximizing their concentration at the target and minimizing toxicity [1]. Specifically, nanoparticles effectively improve the solubility and bioavailability of drugs, prolong the residence time of drugs in the body, and reduce their side effects. Common inorganic nanocarriers include mesoporous silica nanoparticles (MSNs), gold nanoparticles, carbon nanoparticles, magnetic nanoparticles, and quantum dots [2]. Among these nanocarriers, MSNs have received particular attention from many researchers because they can address some of the inherent drawbacks of therapeutics, including limited bioavailability, short cycle life, and adverse biodistribution [3]. In the field of drug delivery, MSNs have many advantages: (1) as micro-reservoirs, they have good ability to contain guest molecules and can release the cargo carried under physiological conditions; (2) they have adjustable pore sizes, so they can be loaded with different carriers; (3) they can achieve targeted delivery and controlled release through surface modification; (4) the combination of loading magnetic and fluorescent compounds can realize the dual functions of drug delivery and biological imaging at the same time [4,5,6,7]. MSN is a type of nanomaterial that is easy to modify and can be combined with other materials to form nanocarriers with multiple functions, such as surface modification. Biomedical applications of MSNs include improving drug solubility, as vectors for controlled/targeted transport, as carriers for diagnostics, and as antigen carriers and adjuvants for vaccine delivery.

The use of poly-dopamine (PDA) to modify material surface properties offers new possibilities for designing nano-carriers [8]. Using PDA to modify the surface properties of nanoparticles is more advantageous than using other surface functional materials. The synthesis of PDA requires simple and gentle, due to its unique adhesion properties, no organic solvents are required, and the synthesis of PDA core-shell nanoparticles can be completed by stirring dopamine hydrochloride in Tris-HCl buffer solution at pH 8.5 [9]. In addition, a variety of different PDA surface modifications can be designed by adjusting basic parameters such as pH, temperature, dopamine concentration, oxidant, and reaction time [10]. Second, the drug loading capacity of nanoparticles can be significantly improved by modification of PDAs, which have rich catechol/quinone structures that bind functional molecules to nanoparticles through physical bonding (π-π or hydrogen bonding) or chemical bonding (Michael addition or Schiff base reaction) [11]. In addition, secondary modifications such as polyethylene glycolation can be achieved with the help of PDAs, as it can react with thiols or amino-containing compounds using Michael addition or a Schiff base [12]. The uniqueness of PDA modifications provides nanoparticles with better hydrophilicity, biocompatibility, and biodegradability, as well as photothermal conversion and reactive oxygen species (ROS) scavenging. This allows PDA-modified nanoparticles to have a variety of functions, from targeting, imaging, and chemical processing (CT) to photodynamic therapy (PDT), photothermal therapy (PTT), tissue regenerative capacity, anti-inflammatory, and antioxidant effects, all of which can play a role in these areas. Therefore, PDA-modified nanoparticles are widely used in cancer treatment, antibacterial applications, therapeutic diagnosis, tissue repair, and other aspects [13,14,15,16].

There are different types of cells in the liver, namely, hepatocytes, Kupfer cells, hepatic stellate cells, and sinusoidal endothelial cells, among others. There are different types of receptors on the surface of these cells, and by binding the drug to a targeted portion that matches the receptor on the surface of hepatocytes, it is possible to precisely target the drug to the liver. Liver-targeted drug delivery systems (HTDDS) can be achieved using a variety of nanocarriers. HTDDS can not only distribute drugs to the liver but also improve the bioavailability of drugs and reduce adverse reactions to drugs [17]. ASGPR receptors are mainly expressed by hepatocytes and are less distributed in extrahepatic cells [18]. It is one of the most studied targets for selective delivery of anti-cancer drugs to hepatocellular carcinoma (HCC) [19]. ASGPR binds to galactose or galactosamine with high affinity and can also be combined with carbohydrates including glucose and polysaccharides and polymers with sugar residues. Thus, galactose, lactose, lactoferrin moiety, and a variety of carbohydrates with repeating galactose or glucose units are used as liver-targeted drug delivery systems [20].

Aspirin (Asp) is a nonsteroidal anti-inflammatory drug that can reduce prostaglandin biosynthesis by inhibiting the activity of COX1 and COX2 (cyclooxygenase1 and cyclooxygenase2). Compared to other NSAIDs, low-dose aspirin (75–100 mg) exerts an antiplatelet drug effect by covalently modifying COX1 expressed in mature platelets. High doses of aspirin (650–1300 mg) can cause COX2 acetylation in inflammatory cells, exerting analgesic and anti-inflammatory effects [21]. There is experimental and clinical evidence that aspirin also has the characteristics of chemoprevention and chemotherapy for cancer. Epidemiological studies in both the general population and high-risk groups have shown a correlation between regular aspirin use and a decrease in the incidence of HCC. Furthermore, regular aspirin use among HCC patients has been found to effectively reduce both recurrence and mortality rates [22]. In addition, preliminary preclinical studies have shown that aspirin’s mechanism of action on HCC may be related to its antiplatelet and anti-inflammatory activities [23,24].

Based on this, MSNs were prepared by the “one-pot two-phase layering method”, functionalized and modified, and the “gated switch” PDA was coated on the surface of MSNs. Then the secondary reactivity of polydopamine was used to connect the target ligand Gal to the surface of nanoparticles, details are shown in Figure 1. Aspirin was selected as a model drug to construct a nano-drug with pH sensitivity and targeting, as shown in Figure 2. The relevant characteristics, biological safety, drug release mechanism, and pharmacodynamic effect on liver cancer cells were preliminarily studied, and analyzed from quality evaluation to pharmacodynamic evaluation, from in vitro to in vivo, to provide a reference for the research of nanomedicine based on secondary modification of polydopamine.

## 2. Results and Discussions

### 2.1. Synthesis and Characterization of MSN-PDA-Gal and Asp@MSN-PDA-Gal

As shown in Figure 1, SEM revealed that the MSN nanoparticles are spherical and exhibit a uniform particle size distribution. TEM showed that the MSN nanoparticles are spherical with clear and uniform mesoporous channels on the surface and a uniform particle size, which is consistent with the SEM results. After modification, MSN-PDA and MSN-PDA-Gal nanoparticles with a uniform particle size and complete morphology were obtained. The particle size distributions of MSN, MSN-PDA, and MSN-PDA-Gal nanoparticles are 143.3 nm, 166.2 nm, and 182.6 nm, respectively (Figure 2). The particle size of the nanoparticles gradually increases with the modification process. The Zeta potential of nanoparticles can reflect the macroscopic changes on the surface of nanoparticles after modification. From Figure 2D, it can be seen that the average surface charge of MSN is −2.9 mV. The negative potential of MSN can be attributed to the existence of a large number of silanol groups on the surface of silica. After surface modification with PDA, the value is reduced to 6.75 mV. This change can be attributed to the deprotonation of phenolic hydroxyl groups of PDA in neutral pH. After functionalization with Gal, the potential of NPs is 12.2 mV. This may be due to the deprotonation of hydroxyl groups in the Gal structure under neutral pH.

The BET nitrogen adsorption/desorption isotherm and BJH pore size distribution of MSNs are shown in Figure 3A,B, and the mesoporous structure of MSNs is an ordered typical type IV isotherm. The specific surface area of MSN was (313.77 m^2^/g), the pore size was (9.84 nm), and the pore volume was (0.772 cm^3^/g).

X-ray photoelectron spectroscopy (XPS) is a technique for analyzing surface elements, the sampling depth is generally between 2–5 nm, so XPS is used to analyze the surface elements of three nanoparticles MSNs, MSN-PDA, Gal-PDA-MSNs, and qualitatively verify the coating of PDA coating and the modification of Gal. The characterization results are shown in Figure 3C–E, MSNs do not contain N elements, but the air contains trace N elements, resulting in N1s high-resolution energy spectrum showing a corresponding peak, and after adding PDA on the surface of nanoparticles, two corresponding peaks of N elements appear on the surface of nanoparticles, which directly proves the successful coating of PDA on the surface of nanoparticles, and after further grafting Gal, the peak height increases significantly because Gal contains nitrogen, and the increase in nitrogen content proves that Gal is modified on the surface of nanoparticles.

### 2.2. In Vitro Release Behavior Studies

To confirm the potential of PDAs as gatekeepers and pH responsiveness of PDA-MSN nanocarriers. In vitro, drug release evaluation experiments were carried out on the prepared MSN@Asp, PDA-MSN@Asp, and Gal-PDA-MSN@Asp. The normal blood pH in the in vivo environment is about 7.4, and the pH value in tumor cells is 4.5–5.5, considering the above factors, pH 7.4 and pH 5.2 were selected for in vitro release experiments.

The linear range examination results in Figure 4E show a good linear relationship (R^2^ = 0.9987) between the concentration of aspirin in the range of 10–60 μg/mL and its absorbance. The standard fitting curve for the exclusivity of aspirin concentration and absorbance is Y = 0.0006x + 0.0197.

Table 1 shows the drug loading rates of MSN at different drug ratios. Among the three regimens, regimen C had the highest drug loading rate of 21.42% and encapsulation rate of 10.71% and was therefore chosen as the loading condition.

From the release curves of Figure 4A,B, it is clear that the pH of different release media has a significant effect on the ASP release of different drug-loaded nanoparticles. When there is no PDA wrapping, the release curves of the three nano drugs at different pH values are not much different, all of which are manifested as faster aspirin release within the first 7 h, due to the large concentration difference between the drugs after the carrier loading and the release medium, so that the release rate of the drug is faster, but over time, the concentration difference will gradually become smaller, so that the release rate will also slowly slow down, and the final concentration difference is close to none, and the release amount also reaches the maximum. The cumulative drug release curves of PDA-MSN@Asp and Gal-PDA-MSN@Asp were significantly different in different pH values. The drug release was slow when the pH value was 7.4 under normal physiological conditions (such as interstitial fluid or blood), and the release characteristics of first sudden release and then sustained release when the pH value was 5.2 in a more acidic environment. In addition, the release curve of Gal-PDA-MSN@Asp is smoother than that of PDA-MSN@Asp. This is because the PDA coating is easy to crack in an acidic environment so that the drug loaded in the MSN can be released, and after Gal modification, the Gal on the surface of the nanoparticles spatially prevents the lysis of the PDA to a certain extent, thereby delaying the drug release, proving that Gal is bound to the surface of the PDA coating. The two release curves can also illustrate the pH-responsive release of PDA coatings to nanoparticles, that is, drug release is slow in neutral environments and more rapid in acidic conditions.

### 2.3. Preliminary Safety Evaluation of Mesoporous Silica Nanocarriers

It has good biosafety, which is the basis for nano-carriers to play a better role in the clinic. Therefore, three kinds of nano-carriers were prepared for preliminary safety evaluation in vivo and in vitro. Firstly, HepG2 human hepatoma cells were selected to investigate the effects of three kinds of nanoparticles on cell activity. Since the therapeutic effect of nano-drugs is achieved by intravenous injection, the hemolysis of nanoparticles into blood circulation was also investigated. In addition, the acute toxicity of nanoparticles in mice was further investigated, including body weight changes, blood routine examination indexes, liver and kidney function, and histopathological changes of main organs.

The cell biocompatibility of MSN, PDA-MSN, and Gal-PDA-MSN nanocarriers was evaluated by the CCK-8 method. The cell survival rate of Figure 4C showed that after 24 h normal culture, when the nanocarrier concentration ≤ 400 μg/mL, the survival rate of HepG2 cells was higher than 80%, and no obvious toxic effect was shown. The concentration of the three nanocarriers was ≤100 μg/mL and had no obvious toxic effect on HepG2 cells, and the survival rate was above 90%. With the increase of the concentration of the carrier material, when the nanocarrier reaches 100 μg/mL, the survival rate of cells decreases but remains above 80%.

As can be seen from Figure 4D, when the sample concentration is less than 200 μg/mL, the hemolysis rates of MSN, PDA-MSN, and Gal-PDA-MSN are all less than 2%, indicating that in this concentration range, the three nano-carriers will not cause hemolysis and have low toxicity. When the concentration range is 200–1000 μg/mL, it still meets the hemolysis test requirements of medical materials. When the concentration of nano-carrier continued to increase, the hemolysis rate of MSN coated with PDA was lower than that of uncoated MSNs, indicating that the coating of PDA increased the blood compatibility of MSN. This may be because the addition of PDA makes MSN more hydrophilic.

Compared with the normal control group, the organ index of mice did not change significantly due to caudal vein injection of nano-carriers Figure 5A–D shows the effect of injection of different types of MSNs on the blood routine of mice, and it can be seen that compared with the normal control group, there are no significant changes in the blood routine RBC, WBC, PLT and HGB of mice. Figure 5E–H shows the changes in liver and kidney functions of mice after injection of different types of MSNs; it can be seen that compared with the normal control group, the three nanocarriers did not cause significant changes in blood-liver function (ALT and AST) and renal function (BUN and CR).

As shown in (Figure 6A), the liver structure of each group is normal, the cells are arranged neatly and tightly, the cytoplasm is uniform, and the central vein is the center of the single row radially arranged. Compared with the normal control group, injection of different types of MSNs did not cause significant changes in liver pathology in mice. The cardiomyocytes of each group had a clear texture, reasonable distribution, and no obvious pathological changes. Compared with the normal control group, injection of different types of MSNs did not cause significant changes in the pathology of cardiac disease in mice (Figure 6B). The structure of glomeruli and nephrostomy can be clearly and unambiguously observed in each group of sections, without the infiltration of inflammatory cells. Injection of different types of MSNs did not cause significant changes in kidney pathology in mice compared to normal controls (Figure 6C). The results of pathological section staining showed that the spleen of the four groups did not have acute toxicological changes, such as inflammatory response, cell edema, and steatosis. Compared with the normal control group, injection of different types of MSNs did not cause significant changes in the pathology of the spleen of mice (Figure 6D). Each group of sections had a complete alveolar structure, the alveolar wall was a monolayer of epithelial cells, and there was no foreign body exudation in the alveolar cavity. Compared with the normal control group, injection of different types of MSNs did not cause significant changes in lung pathology in mice (Figure 6E).

### 2.4. Pharmacodynamic Evaluation of HepG2 Cells by Aspirin-Loaded Nanoparticles

The inhibitory effect of aspirin and aspirin-loaded nanoparticles on the proliferation of HepG2 cells was evaluated by CCK-8 methods. The results of Figure 7A–E showed that both commercial drugs and aspirin-loaded nanoparticles had better inhibitory effect on the growth of HepG2 cells. Compared with Asp, Asp-loaded nanoparticles have lower IC50 value and stronger cytotoxicity. The reason may be that nano-carriers improve the solubility of Asp, increase cell uptake efficiency, and make more drugs enter cells. In addition, the IC50 value of Gal-modified nanoparticles is the lowest, which may be due to the active targeting of Gal-modified MSNs to HepG2 human hepatoma cells and more entry into HepG2 cells, thus enhancing the efficacy.

The effect of Asp-supported nanoparticles on the migration ability of HepG2 cells was studied, and scratch experiments were used to detect them. The results of Figure 8 and Figure 9A show that the healing rate of the Control group within 48 h is higher than that of the experimental group. Compared with MSNs that were not actively targeted modifications, Gal-modified MSNs showed a stronger inhibitory effect on cell migration. There were significant differences between the Gal-PDA-MSN@Asp group and the Asp group. The reason may be that the MSNs modified by Gal are actively targeted and enter more HepG2 cells, thereby enhancing efficacy.

The data of flow cytometry were processed by Flowjo 10.6.2 software. Results, as shown in Figure 9B,C, compared with the blank control group, free coumarin group (C6), and three kinds of nanoparticles group, showed stronger fluorescence intensity and obvious shift, especially the fluorescence intensity of three kinds of nanoparticles group was more than 1 × 10^4^. It was found that the fluorescence intensity of Gal-modified nanoparticles in HepG2 cells was significantly higher than that in the PDA-MSN group and MSN group. The average fluorescence intensity was analyzed by Flowjo 10.6.2 software Figure 9B. The results showed that the fluorescence intensity of coumarin nanoparticles modified by Gal was about 6 × 10^4^ in HepG2 cells, which was about double that of the MSN group and 1.5-times that of the PDA-MSN group, indicating that active targeting modification of Gal promoted the uptake of coumarin nanoparticles by HepG2 cells. The quantitative uptake results also showed that the uptake of PDA-modified nanoparticles by HepG2 cells was better than that of MSN nanoparticles. Some studies have shown that it may be because the PDA coating brings better hydrophilicity to nanoparticles, which makes HepG2 cells uptake more PDA-modified nanoparticles [25].

## 3. Materials and Methods

### 3.1. Chemicals

The chemical reagents required for this experiment are provided by the supplier and used according to the usage specifications. Dopamine hydrochloride, triethanolamine (TEA), Tris buffer (pH = 8.5), D-(+)- galactosamine hydrochloride, and ether were all purchased from McLean Technology Co., Ltd. (Shanghai, China). Anhydrous ethanol, methanol, and cyclohexane were all purchased from Tianjin Damao Reagent Factory (Tianjin, China). Tetraethyl orthosilicate (TEOS) and cetyltrimethyl ammonium chloride (CTAC) were purchased from Sigma-Aldrich Trading Co., Ltd. (Shanghai, China). Unless otherwise specified, all chemical reagents are of analytical grade and used without other treatment.

### 3.2. Animals

Twenty-four 5-week-old KM mice with no specific pathogen level were purchased from Slack Jingda Experimental Animal Co., Ltd. (Changsha, China). (License number: SYXK (Guangdong) 2022-0125), and the quality certificate number of experimental animals was NO.110727201003661. The experimental animals are kept in the pathogen-free laboratory of the Experimental Animal Center of Guangdong Pharmaceutical University. This experiment was approved by the Experimental Animal Ethics Committee of Guangdong Pharmaceutical University and strictly followed the requirements of the Guidelines for Ethical Review of Experimental Animal Welfare (GB/T35892-2018) to fully protect the welfare of experimental animals.

### 3.3. Synthesis of MSN, PDA-MSN, and Gal-PDA-MSN

A mixture of 36 mL of distilled water, 24 mL (25 wt%) of CTAC solution, and 0.18 g of TEA was stirred at 70 °C for 1 h at a stirring rate of 150 rpm/min. Subsequently, 20 mL of TEOS and 2 mL of cyclohexane were added to react. The reaction temperature was set to 70 °C with a magnetic stirring rate of 150 rpm/min, and the reaction time was 24 h. Following the reaction, a high-speed centrifuge was used to spin at 13,000 rpm/min for 20 min to obtain a white precipitate. The crude product was washed three times with ethanol to eliminate any residual reactants. The collected products were washed twice with acetone at an oil bath temperature of 50 °C and continuously stirred for 12 h to remove the template CTAC. Subsequently, the products were washed twice with ethanol to eliminate any residual acetone. Finally, the resulting product was freeze-dried to obtain a powdered MSN product.

MSN (50 mg) was dispersed in 50 mL Tris-HCl buffer (pH 8.5, 10 mmol/L) using ultrasound. Then, dopamine hydrochloride (25 mg) was added and stirred for 20 min. Subsequently, 2 mL of H_2_O_2_ (30 wt%) was added, and stirring was continued for 3 h in the dark. After the reaction, the solid product of PDA-MSNs coated with PDA was centrifuged (12,000 rpm, 10 min) and washed with water to remove unpolymerized dopamine. PDA-MSNs are stored in the refrigerator at 4 °C for later use.

The PDA-MSNs (20 mg) were dispersed in 20 mL Tris-HCl buffer (pH 8.5, 10 mmol/L) using ultrasound. Galactosamine hydrochloride (40 mg) was then added, and the reaction was stirred for 0.5 h at room temperature. After the reaction, unreacted galactosamine hydrochloride was removed by ultrafiltration centrifugation, resulting in the preparation of Gal-PDA-MSNs.

### 3.4. Loading of Aspirin (Asp)

Dissolve 10 mg of aspirin in 5 mL of PBS buffer, then add 5 mg of MSN and magnetically stir for 12 h. MSN loaded with aspirin was centrifuged and freeze-dried for storage. The absorbance of the centrifuged supernatant at 222 nm was measured using an Ultraviolet spectrophotometer (Max M4, Meigu Molecular Instruments Co., Ltd., Shanghai, China). The content of Asp in the supernatant was calculated using the standard curve method. The loading number of aspirin in MSN@Asp was determined by the difference in aspirin content before and after.

Replace MSN with MSN@Asp, and prepare PDA-MSN@Asp and Gal-PDA-MSN@Asp according to Section 2.3.
(1)$$entrapment\;efficiency(EE)=\frac{Asp\;total\;amount-Amount\;of\;Asp\;in\;supernatant}{Total\;amount\;of\;nano-drugs}*100\%$$
(2)$$loading\;capacity\left(LC\right)=\frac{Asp\;total\;amount-Amount\;of\;Asp\;in\;supernatant}{Total\;Asp\;dosage}*100\%$$

### 3.5. Drug Release Experiment of Drug-Loaded Nanoparticles In Vitro

The three nano-drugs were dispersed in 5 mL PBS buffer solution, then the mixture was transferred to 3500 Da dialysis bags, and then placed in 30 mL PBS buffer solution containing 1% SDS (pH = 7.4, 5.2), and slowly oscillated in a constant temperature oscillator at 37 °C and 100 rpm. At specific time points (1 h, 2 h, 3 h, 4 h, 12 h, 24 h, 48 h, and 72 h), 2 mL of precipitated solution was taken out from the outside of the dialysis bag, and then PBS buffer with the same volume was added into the dialysis bag. After the solution was diluted, the concentration of Asp was determined by ultraviolet-visible spectrophotometer at 222 nm, and the content of Asp in each group was calculated according to the standard curve equation, and the cumulative drug release curve was drawn for comparison. Three groups of parallel experiments were set up for each sample, and the results were averaged. The cumulative release rate is calculated by Formula (3):(3)$$Q=\frac{V\sum_{1}^{n-1}C_{i}+V_{0}C_{n}}{M}*100\%$$

*Q*: cumulative release rate of aspirin (%); *V*: sampling volume at a time (mL); *C_i_*: the concentration of aspirin at the first sampling (mg/mL); *V*_0_: Total volume of buffer in dialysis bag (mL); *C_n_*: the concentration of aspirin at the nth sampling (mg/mL); *M*: total dosage of drug-carrying system (mg).

### 3.6. Characterization of Nanoparticles

The size distribution and zeta potential of the prepared nanoparticles were measured by a laser particle size analyzer (Delsa, Beckman Technology Co., Ltd., Durham, NC, USA).

The morphology of the prepared nanoparticles was observed by scanning electron microscope (XFlash 6130, Carl Zeiss, Oberkochen, Germany) and transmission electron microscope (Tecnai G2 F20, Thermo Fisher Scientific, Waltham, MA, USA). A small number of MSNs, MSN-PDA, and Gal-PDA-MSNs powders were uniformly sprayed on the sample base affixed with double-sided conductive adhesive. The sample base was placed in the SEM instrument, and the optimal magnification and observation position were used to take surface morphology shots. A small amount of MSNs, MSN-PDA, and Gal-PDA-MSNs powders were dispersed uniformly in double-distilled water to obtain a sample suspension of moderate concentration, and a drop of about 8 μL of the sample suspension was suspended on the copper grid of the electron microscope, and after the solvent evaporated and dried up, the sample suspension was placed under the TEM to photograph its morphology using the optimal observation magnification and position.

The micropore structure and pore size distribution of the prepared nanoparticles were determined by nitrogen adsorption experiment (asap2460, American Mack Instruments Co., Ltd., Colonial Heights, VA, USA).

### 3.7. Preliminary Safety Evaluation

The human hepatoblastoma cell line (HepG2) was obtained from the cell bank of the China Academy of Sciences and cryopreserved in liquid nitrogen. The cells were cultured in DMEM medium supplemented with 10% fetal bovine serum, 100 units/mL of penicillin, and 100 mg/mL of streptomycin at 37 °C in a 5% carbon dioxide incubator.

After the in vitro cultured HepG2 cells covered the bottom 80% of the culture dish, the cells were inoculated in 96-well plates according to 8000–10,000 cells per well, and the cell status was observed using a microscope. After approximately 12 h of culture, 5, 10, 25, 50, 100, 200, and 400 μg/mL of MSN, PDA-MSN, and Gal-PDA-MSN nanomaterials were administered. Three replicate wells were set up for different concentrations of each nanomaterial. A blank group (medium without cells and drug to be tested, CCK8) and a positive control group (medium containing cells, CCK8, no drug to be tested) were also set up. The incubation was continued for 24 h after the administration of nanomedicine, then the supernatant was removed and a solution of CCK8 at a concentration of 10% configured with DMEM medium was added to each well and placed in an incubator to continue incubation for 1 h before terminating the incubation. The OD value of each well was measured at 450 nm on an enzyme marker and the survival rate was calculated according to the following formula.
(4)$$Cell\;survival\;rate=\frac{A_{s}-A_{b}}{A_{c}-A_{b}}*100\%$$

*A_s_*: absorbance of the experimental hole (culture medium containing cells, CCK-8, sample to be tested); *A_c_*: absorbance of control well (medium containing cells, CCK-8, no sample to be tested); *A_b_*: absorbance of the blank hole (culture medium without cells and samples to be tested, CCK-8).

One mL of fresh rabbit blood was taken and centrifuged at 3000 rpm/min for 10 min to collect the erythrocytes. The erythrocytes were washed with saline until the supernatant was completely colorless, and then the erythrocytes were subsequently reconstituted into 2% erythrocyte suspension with saline. The absorbance was measured by UV spectrophotometry using saline as the negative control and deionized water as a positive control. The nanoparticles were diluted into samples with concentrations of 50, 500, 1000, and 1500 μg/mL with saline, and 2 mL of the sample solution was mixed with 2 mL of 2% erythrocyte solution. After mixing, it was immediately placed in a thermostat for incubation at 37 °C ± 0.5 °C. The absorbance was measured at 540 nm by centrifugation of the supernatant solution after 4 h.

After a week of adaptive feeding, KM mice were randomly divided into four groups, with six mice in each group (half male and half female). Then, the control group was injected with 100 μL of normal saline through the tail vein, and the experimental group was injected with 100 μL of MSN, PDA-MSN, and Gal-PDA-MSN dispersion (dosage: 600 mg/kg), respectively. Then, the body weight was recorded, and the spirits and activities of the mice were observed every day. After feeding for 7 days under normal conditions, the mice were fasted for 12 h, anesthetized with ether, and blood was taken from the orbit. The mice were dissected and the main organs, such as the heart, liver, spleen, lung, and kidney, were taken out and weighed to calculate the organ index. After sampling, the tissue was stored at −80 °C for subsequent detection and analysis.

### 3.8. In Vitro Cellular Uptake

Fluorescent probe-labeled nanoparticles (C6-NPs) were prepared by replacing Asp with an equal amount of coumarin-6 (C6) under light-avoidance conditions according to Section 2.4. HepG2 cells were seeded in 6-well plates at 1 × 10^6^ cells/well and incubated for more than 12 h until 80% cell-adherent coverage was achieved, the original medium was aspirated and the wells were washed with PBS. The C6 solution and C6-NPs were diluted with a serum-free medium so that the fluorescein concentration was 1 μg/mL for both. After the cells in each group were given C6 and C6-NPs, they were incubated at a temperature of 37 °C for 4 h. At the end of the incubation, the fluorescein-containing medium was removed, and the cells were washed with PBS 2 times. Cells were digested with trypsin for 4 min and blown into cell suspension, centrifuged at 1000 rpm/min for 3 min, and then collected and washed twice with PBS. The cells were then fixed with 4% paraformaldehyde for 15 min and collected by centrifugation. Finally, the cells were resuspended with PBS, and 500 μL was placed in a flow tube, and the uptake of coumarin by HepG2 cells was detected using flow cytometry.

### 3.9. Anticancer Effect In Vitro

HepG2 human hepatocarcinoma cells were selected as the experimental object, and commercial aspirin (Asp) was used as the positive control. The inhibitory abilities of Asp, MSN@Asp, PDA-MSN@Asp, and Gal-PDA-MSN@Asp were compared. 10,000 cells per well were seeded in a 96-well plate, and the cell state was observed by a microscope. After about 12 h of culture, four groups of Asp, MSN@Asp, PDA-MSN@Asp, and Gal-PDA-MSN@Asp were set up according to the experimental needs, with 5 wells in each group. Aspirin with the final concentration of 0.1, 1, 5, 10, and 20 mM was given in the form of liquid exchange. Through equivalent conversion, the content of Asp in free drug and nano-drug is ensured to be the same. After the drug was added, the culture was continued for 24 h, and the 10% CCK8 solution prepared with DMEM medium was added by changing the liquid, placed in an incubator, and continued to incubate for 1 h, then the culture was terminated. Measure the OD value of each orifice at 450 nm on an enzyme-labelled instrument, and calculate the cell inhibition rate according to the following formula:(5)$$Cell\;inhibition\;rate=\frac{A_{c}-A_{s}}{A_{c}-A_{b}}*100\%$$

*A_s_*: absorbance of the experimental hole (culture medium containing cells, CCK-8, a drug to be tested); *A_c_*: absorbance of control well (medium containing cells, CCK-8, no drug to be tested); *A_b_*: absorbance of the blank hole (culture medium without cells and drugs to be tested, CCK-8).

Cell migration experiment. Three horizontal lines with equal spacing (about 0.5 cm) were horizontally drawn on the bottom of the 6-well plate with a marker. Cells in the logarithmic growth stage were digested into single-cell suspension by trypsin, diluted to 1 × 10^6^ cells/well, then inoculated into 6-well plates respectively, and 2 mL of DMEM medium was added to each well, and finally put into an incubator with 37 °C and 5% CO_2_ for culture. The negative control group and four experimental groups (Asp, MSN@Asp, PDA-MSN@Asp, Gal-PDA-MSN@Asp) were set up, and each group was set up with three multiple holes. When the cell fusion degree was close to 100%, three vertical lines with equal spacing were drawn on the bottom of the 6-well plate with a sterile gun head of 100 μL, which are perpendicular to the horizontal lines drawn by the marker. The culture medium was removed, cells were washed with PBS buffer twice to remove the scratched cells, cell spacing was observed under the microscope, and photos were taken. Serum-free basic medium was added, the above was repeated after 48 h, and the scratch spacing was measured. Scratch healing rate = (scratch spacing before healing-scratch spacing after healing)/scratch spacing before healing × 100%.

### 3.10. Statistical Analysis

In this experiment, the data were analyzed and plotted by GraphPad Prism 9 software, the two groups were compared by *t*-test, and more than two groups were analyzed by One-way ANOVA. When *p* < 0.05, the difference was significant, and the data results were expressed by mean ± SEM.

## 4. Conclusions

In this study, PDA-modified and Gal-modified mesoporous silicon nanoparticles were constructed using inorganic nanomaterials, and Asp was loaded on the nanoparticles to be used for targeted therapy of liver cancer. The results show that the drug-loaded nanoparticles have a liver-targeting effect and pH-response-release effect, which can accurately act on liver cancer cells and avoid any effects on normal tissues. We preliminarily studied the related properties, biological safety, drug release mechanism, and anti-tumor efficacy (in vivo and in vivo) of Gal-PDA-MSN@Asp, and analyzed it from several aspects including quality evaluation and anti-tumor efficacy, both in vitro to in vivo, to provide references for human body and in vivo evaluation. Gal modification enables nanoparticles to target the salivary acid soup protein receptor on HepG2 to achieve a liver-targeting effect, while PDA modified on the MSN surface has the effect of pH response release, which can release Asp in the acidic microenvironment of a tumor. Compared with aspirin administration alone, Gal-PDA-MSN@AspPDA has a better inhibitory effect on liver cancer cells. In this study, the MSN surface modification strategy is simple, universal, and results in nano-drugs with new surface properties; these can be used as potential carrier platforms for targeted nano-drugs, providing a valuable reference for future targeted drug delivery systems developed according to the characteristics of the tumor microenvironment.

## Data Availability

All data supporting the findings of this study are available in the paper.

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
