# Peer review of "Mesoporous Silicon Nanoparticles with Liver-Targeting and pH-Response-Release Function Are Used for Targeted Drug Delivery in Liver Cancer Treatment"

_ijms, 2024, doi:10.3390/ijms25052525_

Round 1
Reviewer 1 Report
Comments and Suggestions for Authors
The manuscript about “ Mesoporous silicon nanoparticles with liver targeting and pH 2 response release function are used for targeted drug delivery of 3 liver cancer” is scientifically sound and can be accepted after minor revisions suggested below
(1) Can the author prepare a scheme showing Michael addition reaction between Galactosamine and PDA on MSN to understand the readers clearly about this conjugation mechanism.
(2) Can the author show the absorbance standard curve of aspirin.
(3) What was the loading percentage of aspirin in all three nanodrug formulation as it was not mentioned in section 2.4. The author said 5 mg of MSN was used for loading aspirin but haven’t mentioned about other two nano formulations (MSN-PDA, MSN-PDA-Gal).
(4) Please be consistent with the nomenclature of nanodrug formulation as both the nonmenclature MSN-PDA-Gal or Gal-PDA-MSN is used in section 2.3
(5) Please fix sequence of Formulas
(6) Please correct Line 201 “incubated in an incubator at 37℃ 0.5℃
(6) Please improve the resolution of all the images and figures, even the font size of the labels are very small.
(7). It is advisable to change h to days and y-axis is missing in Fig 4B
Author Response
Dear reviewer,
Thank you very much for your valuable comments on the revision of the manuscript ijms-2868762. The following is my reply to the revised draft.
- Can the author prepare a scheme showing Michael addition reaction between Galactosamine and PDA on MSN to understand the readers clearly about this conjugation mechanism.
(I have supplemented the equation of PDA and Gal reaction in the manuscript.)
- Can the author show the absorbance standard curve of aspirin.
(The standard curve of Asp has been supplemented in Figure 4.)
- What was the loading percentage of aspirin in all three nanodrug formulation as it was not mentioned in section 2.4. The author said 5 mg of MSN was used for loading aspirin but haven’t mentioned about other two nano formulations (MSN-PDA, MSN-PDA-Gal).
(The load percentage of Asp has been supplemented in Table 1. Our loading scheme is to load drugs into MSN first, and then modify Asp-MSN with PDA and Gal. I have made a supplementary explanation in section 2.4.)
- Please be consistent with the nomenclature of nanodrug formulation as both the nonmenclature MSN-PDA-Gal or Gal-PDA-MSN is used in section 2.3.
(Has been revised in the manuscript.)
- Please fix sequence of Formulas.
(Has been revised in the manuscript.)
- Please correct Line 201 “incubated in an incubator at 37℃5℃.
(Has been revised in the manuscript.)
- Please improve the resolution of all the images and figures, even the font size of the labels are very small.
(The figure with higher definition has been used.)
- It is advisable to change h to days and y-axis is missing in Fig 4B.
(If the abscissa is represented by days, it may lead to the appearance of decimals. The abscissa of fig. 4B has been modified.)

Reviewer 2 Report
Comments and Suggestions for Authors
Line 40, reference missing.
Line 50-51, I don't understand this sentence. Can you clarify the meaning, expanding this section? The same for line 54-55. Line 57, what does it mean?
Line 91-92, references? What COX stands for?
Line 96 contains a very interesting fact about the chepreventitative effect of aspirin. It is important to expand this part.
You have said you have determined the loading efficiency, but where are the results? Any further consideration on drug release and toxicity should be carried out just only after adjusting the experiments according to the real amounts of drug.
Line 124, a "." Is missing after the long number.
Line 132, what is the stirring rate?
Line 135 (all the methods) use the past tense. Line 132, how many times were the nanoparticles washed?
Line 142,143 check the syntax.
Line 145, was the pbs with or without calcium and magnesium?
Line 148, which spectrometer did you use?
Line 150, add the formula.
Line 152, space missing before PBS.
Line 159, you have said you have used a calibration curve (which was the range and the R2?).
Line 168, choose a more suitable Title.
Describe how the particles were prepared for imaging.
Paragraph 2.7. How did you grow the cells?
Line 183, what does "parallel multiple holes" mean?
Cck8 at what concentration was it used?
What is a blank? and experimental hole?
Line 196-204: this part must be rephrased because it sounds as someone's notes.
Line 223, were the cells kept at 37C for the staining? How did you make the coumarin nanoparticles?
Line 224: what do you mean with "the medium...was digested with trypsin"? Can you explain the method in detail?
Line 226, check the spacings.
Fig 1 a-c, I can't see the scalebars.
Fig 2a-c, the resolution of the images is too low. Same for Figure 3.
Fig.4 the figures don't show the release profile of nanoparticles, but the release profile of aspirin from nanoparticles. Check the x axis labeling.
Fig6 where are the scale bars? The resolution of the photos is very low.
Fig9D is not needed.
Comments on the Quality of English LanguageExtensive editing of English is needed.
Author Response
Dear reviewer,
Thank you very much for your valuable comments on the revision of the manuscript ijms-2868762. The following is my reply to the revised draft.
Line 40, reference missing.
(References have been supplemented in the manuscript.)
Line 50-51, I don't understand this sentence. Can you clarify the meaning, expanding this section? The same for line 54-55. Line 57, what does it mean?
(I have revised it. I want to explain that MSN is a good skeleton for preparing nano-carriers, because it is a nanomaterial that is easy to modify and transform.)
Line 91-92, references? What COX stands for?
(I have added the meaning of the abbreviation, COX stands for cyclooxygenase.)
Line 96 contains a very interesting fact about the chepreventitative effect of aspirin. It is important to expand this part.
(The chemopreventive effect of Asp on cancer has been expanded in the manuscript.
“There is experimental and clinical evidence that aspirin also has the characteristics of chemoprevention and chemotherapy for cancer. Epidemiological studies in both the general population and high-risk groups have shown a correlation between regular aspirin use and a decrease in the incidence of HCC. Furthermore, regular aspirin use among HCC patients has been found to effectively reduce both recurrence and mortality rates.”)
You have said you have determined the loading efficiency, but where are the results? Any further consideration on drug release and toxicity should be carried out just only after adjusting the experiments according to the real amounts of drug.
(The loading efficiency is 21.42%, which I have added in Table 1 of the manuscript.)
Line 124, a "." Is missing after the long number.
(I have added it in the manuscript.)
Line 132, what is the stirring rate?
(The stirring rate is 150 rpm/min, which I have added in the manuscript.)
Line 135 (all the methods) use the past tense.
(It has been revised in the manuscript.)
Line 132, how many times were the nanoparticles washed?
(The nanoparticles were washed three times and have been supplemented in the manuscript.)
Line 142,143 check the syntax.
(It has been revised in the manuscript.)
Line 145, was the pbs with or without calcium and magnesium?
(The PBS used does not contain calcium and magnesium.)
Line 148, which spectrometer did you use?
(The ultraviolet spectrophotometer model is maxm4, meigu molecular instruments co., ltd. I have added it in section 2.4 of the manuscript.)
Line 150, add the formula.
(Added formula)
Line 152, space missing before PBS.
(I have corrected it in the manuscript.)
Line 159, you have said you have used a calibration curve (which was the range and the R2?).
(I have added the standard curve in figure 4E.)
Line 168, choose a more suitable Title.
(I have revised it in the manuscript. The new title is "characterization of nanoparticles".)
Describe how the particles were prepared for imaging.
(I have added it in the manuscript. “The morphology of the prepared nanoparticles was observed by scanning electron mi-croscope (XFlash 6130, Carl Zeiss, Oberkochen, Germany) and transmission electron mi-croscope (Tecnai G2 F20, Thermo Fisher Scientific, Waltham, MA, USA). A small number of MSNs, MSN-PDA, and Gal-PDA-MSNs powders were uniformly sprayed on the sample base affixed with double-sided conductive adhesive, and the sample base was placed in the SEM instrument, and the optimal magnification and observation position was used to take sur-face morphology shots. A small amount of MSNs, MSN-PDA, and Gal-PDA-MSNs powders were dispersed uniformly in double-distilled water to obtain a sample suspension of mod-erate concentration, and a drop of about 8 μL of the sample suspension was suspended on the copper grid of the electron microscope, and after the solvent evaporated and dried up, the sample suspension was placed under the TEM to photograph its morphology using the optimal observation magnification and position.”
Paragraph 2.7. How did you grow the cells?
(I have supplemented in 2.7 how to cultivate cells. “The human hepatoblastoma cell line (HepG2) was obtained from the cell bank of the China Academy of Sciences and cryopreserved in liquid nitrogen. The cells were cultured in DMEM medium supplemented with 10% fetal bovine serum, 100 units/ml of penicillin, and 100 mg/ml of streptomycin at 37℃ in a 5% carbon dioxide incubator.”)
Line 183, what does "parallel multiple holes" mean?
(My expression here is wrong. I want to explain that the cytotoxicity test of each group is repeated three times, n=3.)
Cck8 at what concentration was it used?
(10% CCK8 solution was prepared with DMEM medium.)
What is a blank? and experimental hole?
(A blank group (medium without cells and drug to be tested, CCK8) and a positive control group (medium containing cells, CCK8, no drug to be tested) were also set up.)
Line 196-204: this part must be rephrased because it sounds as someone's notes.
(I have rewritten it in the manuscript.)
Line 223, were the cells kept at 37C for the staining? How did you make the coumarin nanoparticles?
(Cells were stained at 37℃. “Fluorescent probe-labeled nanoparticles (C6-NPs) were prepared by replacing Asp with an equal amount of coumarin-6 (C6) under light-avoidance conditions according to section 2.4. HepG2 cells were seeded in 6-well plates at 1×106 cells/well and incubated for more than 12 h until 80% cell-adherent coverage was achieved, the original medium was aspirated and the wells were washed with PBS. The C6 solution and C6-NPs were diluted with a serum-free medium so that the fluorescein concentration was 1 μg/ml for both. After the cells in each group were given C6 and C6-NPs, they were incubated at a temperature of 37°C for 4 h.”)
Line 224: what do you mean with "the medium...was digested with trypsin"? Can you explain the method in detail?
(At the end of the incubation, the fluorescein-containing medium was removed, and the cells were washed with PBS 2 times. Cells were digested with trypsin for 4 min and blown into cell suspension, centrifuged at 1000 rpm/min for 3 min, and then collected and washed twice with PBS.)
Line 226, check the spacings.
(I have revised it in the manuscript.)
Fig 1 a-c, I can't see the scalebars.
(I have added it in the manuscript.)
Fig 2a-c, the resolution of the images is too low. Same for Figure 3.
(I have changed the picture with higher resolution in the manuscript.)
Fig.4 the figures don't show the release profile of nanoparticles, but the release profile of aspirin from nanoparticles. Check the x axis labeling.
(I have revised Figure 4 in the manuscript.)
Fig6 where are the scale bars? The resolution of the photos is very low.
(Fig. 6 is a view at a magnification of 200X. I have updated Figure 6 with higher resolution.)
Fig9D is not needed.
(I have removed Figure 9D from the manuscript.)

Round 2
Reviewer 2 Report
Comments and Suggestions for Authors
Fig 6 and 8, I can't see the scalebars. Please, made them visible on the images.
Comments on the Quality of English LanguageMinor English Editing
Author Response
Dear reviewers,
Thank you again for your suggestion. The following is my modification.
I have increased the scale for Figure 6 and Figure 8.